# Cardiac Magnetic Resonance to Detect the Underlying Substrate in Patients with Frequent Idiopathic Ventricular Arrhythmias

**DOI:** 10.3390/diagnostics11061109

**Published:** 2021-06-18

**Authors:** Chrysovalantou Nikolaidou, Christos P. Kotanidis, Rohan Wijesurendra, Joana Leal-Pelado, Konstantinos Kouskouras, Vassilios P. Vassilikos, Haralambos Karvounis, Ntobeko Ntusi, Charalambos Antoniades, Stefan Neubauer, Theodoros D. Karamitsos

**Affiliations:** 1Oxford Centre for Clinical Magnetic Resonance Research, University of Oxford, Oxford OX3 9DU, UK; Chrysovalantou.nikolaidou@cardiov.ox.ac.uk (C.N.); rohan.wijesurendra@cardiov.ox.ac.uk (R.W.); joana.leal@cardiov.ox.ac.uk (J.L.-P.); stefan.neubauer@cardiov.ox.ac.uk (S.N.); 2First Department of Cardiology, AHEPA Hospital, Faculty of Health Sciences, School of Medicine, Aristotle University of Thessaloniki, 546 36 Thessaloniki, Greece; hkarvounis@auth.gr; 3Radcliffe Department of Medicine, Division of Cardiovascular Medicine, University of Oxford, Oxford OX3 9DU, UK; christos.kotanidis@gtc.ox.ac.uk (C.P.K.); charalambos.antoniades@cardiov.ox.ac.uk (C.A.); 4Department of Radiology, AHEPA Hospital, Faculty of Health Sciences, School of Medicine, Aristotle University of Thessaloniki, 546 36 Thessaloniki, Greece; coskou@auth.gr; 5Third Department of Cardiology, Hippokration Hospital, Faculty of Health Sciences, School of Medicine, Aristotle University of Thessaloniki, 546 43 Thessaloniki, Greece; vvassil@auth.gr; 6Department of Medicine, University of Cape Town and Groote Schuur Hospital, 7925 Cape Town, South Africa; Ntobeko.Ntusi@uct.ac.za

**Keywords:** cardiac magnetic resonance, premature ventricular contractions, ventricular arrhythmia, myocardial strain, feature-tracking CMR

## Abstract

Background: A routine diagnostic work-up does not identify structural abnormalities in a substantial proportion of patients with idiopathic ventricular arrhythmias (VAs). We investigated the added value of cardiac magnetic resonance (CMR) imaging in this group of patients. Methods: A single-centre prospective study was undertaken of 72 patients (mean age 46 ± 16 years; 53% females) with frequent premature ventricular contractions (PVCs ≥ 500/24 h) and/or non-sustained ventricular tachycardia (NSVT), an otherwise normal electrocardiogram, normal echocardiography and no coronary artery disease. Results: CMR provided an additional diagnostic yield in 54.2% of patients. The most prevalent diagnosis was previous myocarditis (23.6%) followed by possible PVC-related cardiomyopathy (20.8%), non-ischaemic cardiomyopathy (8.3%) and ischaemic heart disease (1.4%). The predictors of abnormal CMR findings were male gender, age and PVCs/NSVT non-outflow tract-related or with multiple morphologies. Patients with VAs had an impaired peak left ventricular (LV) global radial strain (GRS) compared with the controls (28.88% (IQR: 25.87% to 33.97%) vs. 36.65% (IQR: 33.19% to 40.2%), *p* < 0.001) and a global circumferential strain (GCS) (−17.66% (IQR: −19.62% to −16.23%) vs. −20.66% (IQR: −21.72% to −19.6%), *p* < 0.001). Conclusion: CMR reveals abnormalities in a significant proportion of patients with frequent idiopathic VAs. Male gender, age and non-outflow tract PVC origin can be clinical indicators for CMR referral.

## 1. Introduction

Ventricular arrhythmias (VAs) are frequent in the general population and can present with a wide range of clinical manifestations from benign premature ventricular contractions (PVCs) to life threatening episodes of complex VAs such as ventricular tachycardia or fibrillation [1]. PVCs can be found in 40% to 75% of individuals on 24 to 48 h Holter monitoring and account for approximately 90% of “idiopathic” VAs in patients with structurally normal hearts while non-sustained or sustained ventricular tachycardia or fibrillation are less frequent in this context [2]. VAs can reflect primary electrical disease or may be associated with structural heart disease (SHD), major adverse cardiac events and an increased risk of sudden cardiac death (SCD). The prognostic significance of idiopathic VAs remains unclear and the data regarding the risk of SCD are discordant. However, the association of frequent VAs and potentially reversible arrhythmia-induced cardiomyopathy and systolic dysfunction has been well established [1,3,4]. 

The diagnosis, prognostication and treatment of patients with VAs are challenging. A routine diagnostic workup with transthoracic echocardiography and an assessment for the presence of coronary artery disease (CAD) as suggested by current clinical guidelines [1] cannot recognise focal structural abnormalities or underlying SHD in a substantial proportion of patients [4]. Cardiac magnetic resonance (CMR) provides an excellent assessment of cardiac morphology and function and enables a detailed myocardial tissue characterisation with a high degree of precision. CMR is widely regarded as the gold standard for identifying structural arrhythmogenic substrates in patients with VAs and normal echocardiography [5,6,7,8]. More importantly, myocardial structural abnormalities detected on CMR in patients with idiopathic VAs are associated with an increased risk of arrhythmic events and worse outcomes [4,9,10]. Interestingly, a pattern of ‘ring-like’ mid-myocardial/subepicardial scars on CMR was found to be associated with a high risk of future malignant arrhythmic events [11].

The purpose of this study was to evaluate the added value of CMR including feature-tracking strain CMR (FT-CMR) in identifying underlying SHD in a relatively ‘healthy’ population of patients with VAs but normal echocardiography and testing for CAD.

## 2. Materials and Methods

### 2.1. Study Design and Patient Population

A single-centre, longitudinal and prospective study was undertaken of 72 consecutive patients referred for a CMR scan due to frequent PVCs and/or NSVT, normal findings on echocardiography and no evidence of CAD. The echocardiographic studies were performed by the referring physicians. All participants provided written informed consent and the study protocol was approved by the Ethics Committee of our institution. The CMR findings were compared with those from 72 healthy volunteers matched for age and body surface area (BSA) without a history of PVCs/NSVT or palpitations. The electrocardiogram (ECG) recordings of patients were analysed by an expert electrophysiologist (RW) blinded to the CMR findings to determine the possible arrhythmic focus. 

The inclusion criteria were the presence of frequent PVCs (≥500/24 h) or NSVT on ECG and/or Holter monitoring; otherwise, a normal ECG with no changes suggestive of structural heart disease, normal echocardiography findings and no evidence of CAD on anatomical and/or functional imaging. All echocardiograms were performed by the referring physicians; the assessment of the LV systolic function was performed with the biplane Simpson disk summation method. The exclusion criteria were a known history of cardiac disease (CAD, cardiomyopathy, congenital heart disease, more than mild valvular disease, previous cardiac surgery of any type), any severe systemic disease that could affect the heart, an allergy to gadolinium-based contrast agents, an estimated glomerular filtration rate < 30 mL/min/1.73 m^2^, significant electrolyte abnormalities, any contraindication to the MR environment (e.g., MR-unsafe implants/devices, shrapnel injury), pregnancy and claustrophobia. 

### 2.2. CMR Protocol

The scans were performed on a 1.5 T Magnetom Avanto (Siemens, Erlangen, Germany) scanner using phased-array radiofrequency receiver surface coils and dedicated cardiac software. The images were obtained with breath-holding instructions and ECG gating. A standard cardiac protocol was used including balanced steady state free precession cine images (repetition time (TR) ~45.30 ms, echo time (TE) 1.27 ms, flip angle 55°, field of view (FOV) 360 to 420 mm, base resolution 256 × 200, sequential 7 mm slices with a 3 mm interslice gap, 30 phases per cardiac cycle) of three long-axes, short-axis slices and RV dedicated views and axial RV slices (sequential 6 mm slices with no interslice gap); T2-weighted fat-suppressed turbo spin echo imaging (three short-axis 10 mm slices, TR of 2 RR intervals, TE 50 ms, inversion time 170 ms, FOV 360 to 400 mm, base resolution 192 × 256) and late gadolinium enhancement imaging (gadobutrol 0.1 mmol/kg, TR 700 ms, TE 1.37 ms, flip angle 55°, FOV 380 to 420 mm, base resolution 256 × 232, sequential 7 mm slices with a 3 mm interslice gap). Arrhythmia detection and sorting algorithms were used in patients with occasional PVCs.

In patients with a high burden of PVCs (bigeminy or trigeminy), we administered procainamide on the scanner table to acquire high quality retrospective cine images, which has been shown to be effective and safe for the suppression of PVCs as previously described [12]. More specifically, procainamide hydrochloride (Biocoryl^®^, vial 1 g/10 mL) was administered on the scanner table at intermittent i.v. bolus doses of 50 mg every minute until the suppression of PVCs was achieved or a maximum dose of 10 mg/kg was reached. Procainamide acts mainly by delaying repolarisation and increasing the effective refractory period of atrial and ventricular fibres [13]. The onset of action of procainamide is 10–30 min after i.v. administration with peak concentrations achieved in 15–60 min and a half-life of 2.5–5 h in patients with a normal renal function [14,15]. The ability to suppress automaticity and the quick onset and short duration of procainamide when administered in bolus i.v. doses makes it an ideal antiarrhythmic agent for PVC suppression particularly in patients with a normal or only mildly impaired LV function [16]. During the administration of procainamide to the patients, their blood pressure was measured every minute and there was a continuous monitoring of the heart rate and ECG trace using a magnetic resonance-compatible vital signs monitoring system.

The CMR scans were analysed using dedicated cardiac software (cmr42, Circle Cardiovascular Imaging Inc., Calgary, AB, Canada). The ventricular volumes and mass were quantified from the cine short-axis stack with a manual contouring of the epicardium and endocardium at the end-diastole and end-systole; the left ventricular (LV) papillary muscles were included in the volumes. Different types of non-ischaemic cardiomyopathy (NICM) were diagnosed using the published diagnostic criteria in the absence of other cardiac or systemic diseases that could explain the cardiac appearances [17,18]. A diagnosis of a previous myocardial infarction (MI) was based on the typical subendocardial to transmural pattern of scarring in keeping with a coronary artery territory [19]. Possible previous myocarditis was diagnosed in patients with the typical subepicardial/mid-wall pattern of late gadolinium enhancement (LGE); additional evidence of an acute myocardial oedema on T2-weighted imaging was required for acute myocarditis [20]. Patients with borderline or only a mildly impaired LV systolic function but without LV dilatation or any other imaging evidence suggestive of an NICM were diagnosed with possible PVC-induced cardiomyopathy. PVC-induced cardiomyopathy is a potentially reversible condition in which the LV dysfunction is considered to be due to frequent premature ventricular contractions. The LV systolic function usually improves after PVC suppression [21]. 

The myocardial strain was measured using the feature-tracking CMR technique (FT-CMR) on the SSFP short-axis cine stack and three long-axes. The base was selected as the slice closest to the mitral valve annulus on the short-axis images without through plane distortion from the LV outflow tract; the apical slice was selected as the most apical short-axis slice with clear epicardial and endocardial contours. Left and right ventricular (RV) endocardial and epicardial borders were manually drawn in the end-diastole and were automatically propagated through the cardiac cycle. An analysis of a random set of scans by two of the investigators (CN, JLP; one blinded to the results) showed an excellent reproducibility for all strain variables (r > 0.90). The normal FT-CMR strain values were similar to those derived from speckle-tracking echocardiography (STE), with a good inter-technique agreement for the LV and a moderate intermodality agreement for the RV. Among the strain parameters derived on the FT-CMR, the LV global circumferential strain showed the smallest interobserver and intraobserver variability [22,23,24,25,26]. 

### 2.3. Statistical Analysis

We present continuous variables as means (SD) or medians (25th to 75th percentile), as appropriate. The normality of distribution was tested using the Shapiro–Wilk test. We compared the categorical variables between two or more groups with the chi-squared test and we compared the continuous variables with either a Student’s *t*-test or a Mann–Whitney U-test (for two groups, as appropriate) or by ANOVA (for more groups). We selected the optimum cut-off for the peak LV GRS by identifying the value that maximised Youden’s J statistic. We assessed the independent ability of the peak LV GRS to identify patients using a multivariable logistic regression model adjusted for LVEF, LV end-diastolic volume index, age, sex, BMI, smoking, hypertension and dyslipidaemia. Statistical analyses were performed in the R environment (R version 3.6.0 and R Studio version 1.2.1335). All tests were two-sided and α was set at 0.05. 

## 3. Results

### 3.1. Patient Characteristics and CMR Findings

The main baseline characteristics of the patients and healthy control subjects are shown in Table 1 (see Appendix A). Most of the patients were referred for CMR due to PVCs (n = 56, 77.8%) while 22.2% had NSVT. The average number of PVCs on 24 h Holter monitoring was 15,936 ± 12,894 with 61% of the patients having more than 10,000 PVCs. Compared with the control subjects, the patients with VAs had higher LV volumes and a lower LV and RV ejection fraction, albeit within a normal reference range; the RV volumes were not different. 

CMR detected abnormal findings in 39 of the 72 patients (54.2%) with possible previous myocarditis being the most prevalent diagnosis (n = 17; 23.6%). A significant proportion of the patients (n = 15; 20.8%) with a borderline or only a mildly impaired LV systolic function but without LV dilatation or any other imaging evidence suggestive of an NICM were diagnosed with possible PVC-induced cardiomyopathy, as described above. Six patients (8.3%) were diagnosed with NICMs (four patients with dilated cardiomyopathy, one patient with possible left dominant arrhythmogenic cardiomyopathy and one patient with possible cardiac sarcoidosis) while only one patient had a previous subendocardial infarction (Figure 1).

The differences between patients with a normal CMR scan and those who had abnormal CMR findings are shown in Table 2. The patients with abnormal CMR findings were older and more frequently male and had higher LV volumes and a lower LV and RV ejection fraction compared with patients with a normal CMR scan. As expected, LV LGE was more common in patients with abnormal CMR findings. None of the patients had evidence of RV scarring/fibrosis on LGE or evidence of a myocardial oedema on T2-weighted imaging. The number of PVCs did not differ between the patient groups. Additionally, there was no difference when patients were divided into two groups of those with more or less than 10,000 PVCs/24 h. Although there was a numerical difference for NSVT in patients with an abnormal CMR scan, this did not reach a statistical significance. 

Procainamide was administered intravenously in 29 (40.3%) of the patients on the scanner table prior to CMR scanning. The average dose of procainamide administered was 536 ± 200 mg (range 200–1000 mg). There was a small but statistically significant drop in systolic blood pressure after procainamide administration but no significant drop in diastolic blood pressure or heart rate (see Appendix A). None of the patients developed significant electrocardiographic changes such as a heart block, bradycardia, QRS or QT prolongation or exacerbation of the ventricular arrhythmia. Procainamide successfully suppressed PVCs in 96.6% of the patients (10 patients with complete suppression and 18 with significant reduction, i.e., less than 1 PVC in 10 normal sinus beats enabling the use of arrhythmia detection and sorting algorithms). There was no effect of procainamide in only one patient. The overall image quality and diagnostic capability significantly improved after PVC suppression with procainamide (see Appendix A). 

The majority of the patients (69.2%; 45 out of 65 patients for which PVC morphology was available from the ECG recordings) had PVCs originating from the ventricular outflow tracts; eleven patients had a probable LV origin, two had a presumed RV focus and three patients had two different PVC morphologies (see Appendix A). When the PVC origin was grouped as common (LV and RV outflow tracts) and uncommon (ventricular focus or two different morphologies), it was found to be significantly different between the two groups. Interestingly, the PVC origin remained a significant predictor of the CMR findings with an OR of 5.09 (95% CI: 1.40–22.26, *p* = 0.02) after adjusting for age and number of ectopics. 

### 3.2. Myocardial Strain Findings

An FT-CMR analysis was performed for 67 patients (93.1%). Four patients were excluded due to an impaired image quality from frequent PVCs; the patient with a previous MI was also excluded. Figure 2 shows the peak strain 16-segment maps and strain curves derived from the FT-CMR strain analysis in a patient with normal CMR findings (panels D–F) and a patient with a mid-wall septal fibrosis (panels G–I). 

The strain measurements were compared with those from the seventy-two matched healthy controls. Compared with the control subjects, the patients with VAs had an impaired peak LV global short-axis radial strain (GRS) (28.88% (IQR: 25.87% to 33.97%) vs. 36.65% (IQR: 33.19% to 40.2%), *p* < 0.001) and global circumferential strain (GCS) (−17.66% (IQR: −19.62% to −16.23%) vs. −20.66% (IQR: −21.72% to −19.6%), *p* < 0.001) (Figure 3A). 

The peak LV GRS showed a good diagnostic accuracy in detecting patients from the control subjects (AUC: 0.78 (95% CI: 0.69–0.86), *p* < 0.001) (Figure 3B). Moreover, in a multivariable regression model, the subjects with a low GRS (< 29.91% determined by Youden’s index) had five-fold higher odds of having VAs (OR: 4.99 (95% CI: 1.2–21.95)) after adjusting for LVEF, LV end-diastolic volume index, age, sex, BMI, smoking, hypertension and dyslipidaemia. The peak LV global longitudinal strain (GLS) and RV strain indices were not statistically different between the patients and the control subjects.

### 3.3. Myocardial Strain in Different Patient Groups

Figure 4 shows the peak LV GLS and peak LV GRS strain values in the different patient groups. The peak LV GCS could differentiate patients with previous myocarditis from patients with NICMs. The peak LV GRS could differentiate patients with possible previous myocarditis from patients with NICMs and those with PVC-related cardiomyopathy. No statistically significant difference was found among the other patient groups. Interestingly, the patients with frequent VAs and normal CMR findings had a significantly impaired peak LV GCS and peak LV GRS compared with the control subjects independent of LVEF and the presence of LGE. The findings were also independent of the number of PVCs on 24 h Holter monitoring.

### 3.4. Patient Outcomes

Ten patients with pathological CMR findings (26%; five patients with possible previous myocarditis; four patients with PVC-related cardiomyopathy; one patient with an NICM) underwent catheter ablation for the ventricular ectopy compared with three patients (9%) with normal CMR findings. One patient with an abnormal CMR was offered catheter ablation but refused. The catheter ablations were requested by the referring physicians. The rest of the patients in both groups remained stable with medical treatment. 

## 4. Discussion

The present study showed that (i) CMR could identify underlying heart disease in a substantial number of patients with VAs and normal transthoracic echocardiography, (ii) myocardial deformation indices (peak global GRS and GCS) were more sensitive than the ejection fraction to depict contractile dysfunction in this cohort and (iii) male gender, age and the origin of VAs were independent predictors of abnormal CMR findings (Figure 5). Importantly, our study included a healthier patient cohort compared with previous studies underlying further the value of CMR in the diagnostic work-up of patients with VAs and normal findings on conventional investigations. 

Based on the CMR findings, a definitive diagnosis of underlying SHD was established in 33.4% of patients with frequent PVCs/NSVT, a normal echo and no evidence of CAD. Additionally, approximately one in five patients (20.8%) was found to have a borderline or mildly impaired LV systolic function most likely related to the high ectopic burden, which was not detected on echocardiography. Therefore, CMR provided an additional diagnostic yield in 54.2% of patients considered to have normal hearts. Importantly, CMR, being the gold standard method for the assessment of myocardial volumes and function, demonstrated an impaired systolic function in a proportion of patients who had not been diagnosed on the echocardiogram performed by the referring physicians. 

Our findings were consistent with those from previous CMR studies that showed the additive value of CMR in 31.6% to 78.8% of patients with frequent VAs and normal echocardiography [5,6,7,8,27]. Similar to previous studies, the most common diagnosis was myocarditis, which is associated with a wide spectrum of VAs during the acute or healed phase [28]. However, as expected, there was a much lower prevalence of previous MI because, in contrast to previous studies, all patients were rigorously assessed to exclude CAD. 

Predictors of abnormal CMR findings on the univariate regression analysis were male gender and age, as has been previously shown in patients with idiopathic VAs [5], especially those of a LV origin [10]. Although one study showed that the PVC burden was higher in patients with abnormal CMR findings [7], our study confirmed previous observations that the number of PVCs was not predictive of pathological CMR findings [5,29]. On the contrary, one study in athletes found that those with no scarring on LGE imaging had a higher PVC burden than athletes with LGE findings [30]. This may be explained by the fact that ectopic foci located in the outflow tract of the ventricles may give rise to very frequent PVCs [31]. Moreover, in patients with VAs with a left bundle branch block morphology, the number of PVCs was not predictive of CMR abnormalities in contrast to the presence of VT [32]. In our study, there was a numerical difference for NSVT in patients with an abnormal CMR scan but this did not reach a statistical significance. Non-outflow tract PVCs and NSVT or PVCs with multiple morphologies as deduced from a surface ECG analysis were also predictive of CMR abnormalities. Previous studies have shown that patients with VAs with a presumed LV origin and patients with multiple PVC patterns have a higher prevalence of structural heart disease and/or myocardial fibrosis on CMR [33,34,35] while patients with RVOT arrhythmias have similar CMR findings to normal controls [36]. 

Although EF is the most commonly used measure of systolic function, myocardial strain has emerged as a superior index of systolic performance with an incremental prognostic value [37]. More importantly, myocardial deformation can be more sensitive than EF in detecting incipient contractile dysfunction [38]. Subtle changes in myocardial mechanics have been detected with speckle-tracking echocardiography in patients with frequent VAs and structurally normal hearts [39,40]. Even though the axial spatial and temporal resolution are superior in echocardiography to those in CMR, these are blunted by the low signal-to-noise ratio. Another limitation of 2D echocardiography is that, unlike CMR, the imaging planes may not depict the true apex of the heart, an effect known as foreshortening [41]. We evaluated the peak global LV and RV myocardial strain parameters on the FT-CMR, which have shown a better reproducibility than regional measures [42,43]. The peak global GRS and GCS were impaired in patients with VAs compared with control subjects independent of LVEF, with the peak LV GRS having a good diagnostic accuracy in detecting the patients from the control subjects. A previous study in 42 patients with VAs and structurally normal hearts found abnormal segmental GRS and GCS measures in the basal and middle LV segments and impaired GLS only in the mid-LV segments compared with the healthy control subjects [44]. In our study, the peak LV GLS and RV strain values showed no statistical difference between the patients and the controls. Future studies including a larger cohort of patients may detect differences in these strain measures if they are present. Furthermore, advancements in post-processing software packages or in the temporal resolution of CMR studies may improve the visualisation of the thin-walled RV to generate more reliable strain measurements [23]. 

CMR has an important clinical role in patients with idiopathic VAs. First of all, it can diagnose patients with cardiomyopathy and re-classify their risk of future adverse cardiac events and SCD. Second, it can identify the presence and location of scar tissue (for example, from previous myocarditis) and guide treatment with ablation [4]. Moreover, incipient myocardial dysfunction detected as a borderline LVEF or an impaired myocardial strain on CMR can guide the optimisation of medical treatment or catheter ablation to protect from a further reduction and improve cardiac function [45].

A few limitations of this study should be acknowledged. First, our sample size was relatively small and referral bias could not be avoided. Another limitation of the study was the lack of data from the speckle-tracking strain analysis as echocardiograms were performed by referring physicians so the correlation of the strain measurements on the FT-CMR with STE in our patient cohort was not possible. Furthermore, it should also be noted that arrhythmia-related cardiomyopathy is a clinical diagnosis and requires follow-up and reassessment of the LV systolic function after the suppression of VAs otherwise a mild/early dilated cardiomyopathy phenotype cannot be excluded with certainty. In addition, although we diagnosed possible previous myocarditis in patients with the typical subepicardial/mid-wall pattern of LGE [46], other NICMs with a similar pattern of scarring could not be completely excluded. Moreover, parametric mapping (T1, T2 and ECV mapping) for the assessment of a myocardial oedema and tissue characterisation was not performed as the sequences were not available at our centre. Finally, it should be taken into consideration that, in a few of the cases, the myocardial pathology and/or scarring identified on CMR may be just an incidental finding not directly related to the PVC aetiology [47]. 

## 5. Conclusions

CMR revealed structural abnormalities concealed to conventional diagnostic investigations in a significant proportion of patients with PVCs/NSVT and normal transthoracic echocardiography. Male gender, older age and non-outflow tract PVC origin but not the arrhythmic burden could be used as clinical indicators for referring for a CMR scan. The assessment of myocardial contractility should go beyond the simple measurement of the ejection fraction as the peak LV GRS and the peak LV GCS on the FT-CMR constitute markers of myocardial dysfunction on top and independently of EF.

## Figures and Tables

**Figure 1 diagnostics-11-01109-f001:**
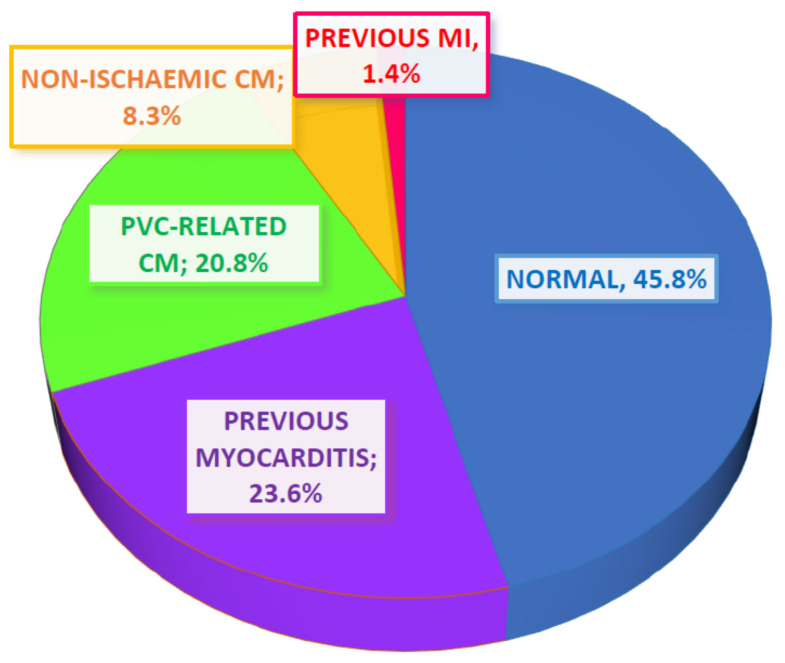
Pie chart showing the main categories of CMR diagnoses within the patient population. Most of the patients had normal CMR findings; myocarditis was the second most common diagnosis while a significant proportion of patients were characterised as having possible PVC-related cardiomyopathy. A small proportion of patients were diagnosed with non-ischaemic cardiomyopathy while only one patient had a myocardial infarction. CM: cardiomyopathy; MI: myocardial infarction; PVC: premature ventricular contractions.

**Figure 2 diagnostics-11-01109-f002:**
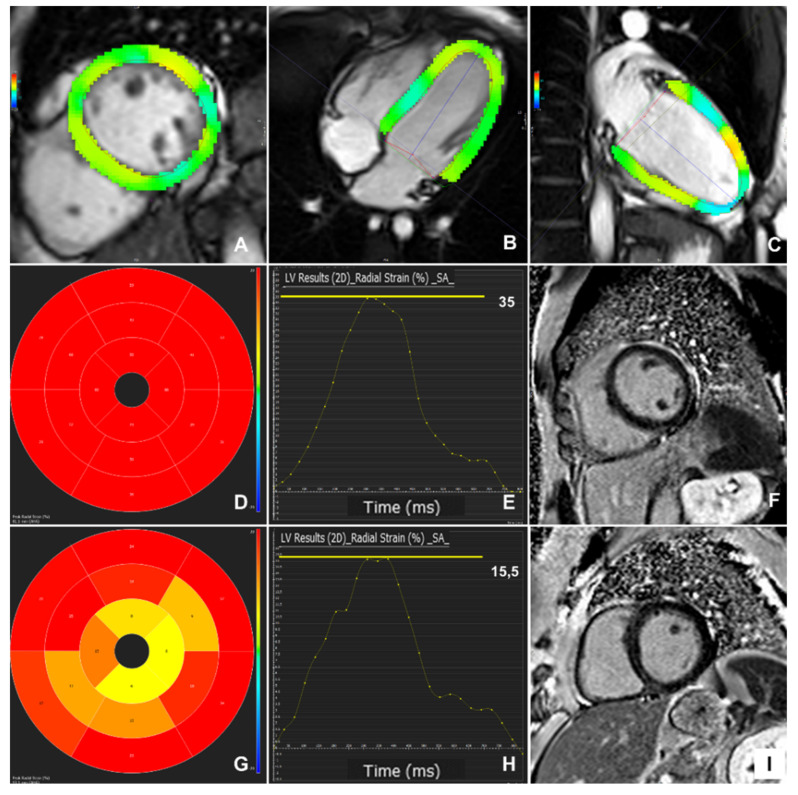
Feature-tracking CMR 2D left ventricular coloured strain analysis on short-axis (**A**) and long-axes (**B**,**C**) SSFP images. Normal peak radial strain 16-segment map (**D**) and strain curve (**E**) from a patient with normal CMR findings and no enhancement on late gadolinium imaging (**F**); impaired peak radial strain 16-segment map (**G**) and strain curve (**H**) in a patient with an enhancement pattern on late gadolinium imaging in keeping with previous myocarditis (**I**).

**Figure 3 diagnostics-11-01109-f003:**
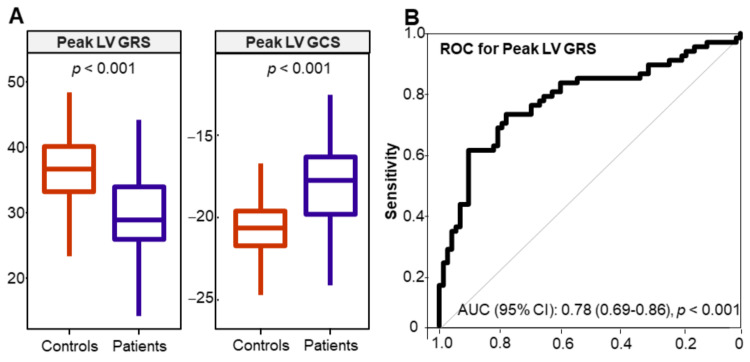
(**A**) Box plots showing the median values of the peak left ventricular global radial strain (LV GRS) and the peak left ventricular global circumferential strain (LV GCS) in control subjects and patients with ventricular arrhythmia. (**B**) Receiver operating characteristic (ROC) curve for the peak left LV GRS; the curve shows a good ability of the peak LV GRS to differentiate the patients from the control subjects. AUC: area under the curve.

**Figure 4 diagnostics-11-01109-f004:**
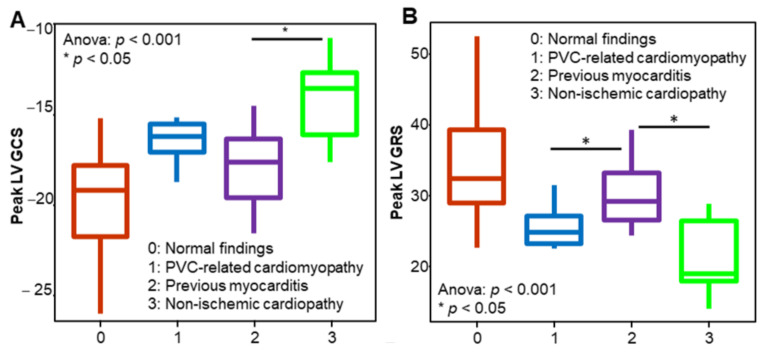
Box plots showing the median values of the peak left ventricular global circumferential strain (LV GCS) (**A**) and the peak left ventricular global radial strain (LV GRS) (**B**) in the different patient groups. Patients with non-ischaemic cardiomyopathy had a significantly impaired peak LV GCS and peak LV GRS compared with patients with previous myocarditis. The peak LV GRS can also differentiate between patients with previous myocarditis and those with premature ventricular contraction (PVC)-related cardiomyopathy. The asterisk indicates *p* < 0.05.

**Figure 5 diagnostics-11-01109-f005:**
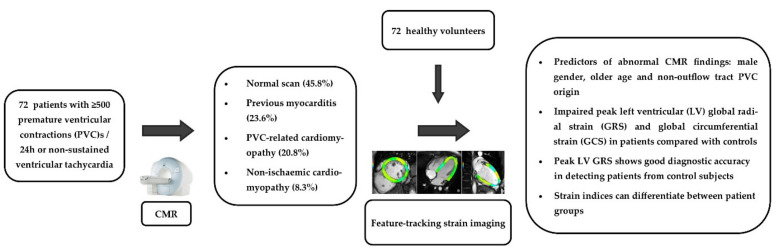
Graphical representation of the study protocol and the main findings.

**Table 1 diagnostics-11-01109-t001:** Baseline clinical characteristics and CMR measurements of patients and matched control subjects.

All Patients (n = 72)		Controls (n = 72)	*p*-Value
Clinical characteristics			
Age (years)	49.5 (36.8, 58.0)	48.5 (40.0, 58.0)	0.701
Males, n (%)	34 (47.2)	34 (47.2)	1.0
BSA (m^2^)	1.9 ± 0.2	1.9 ± 0.2	0.809
Smoking, n (%) *	15 (24.2)	2 (2.8)	**0.001**
Hypertension, n (%) ^†^	12 (17.4)	7 (9.7)	0.277
Hyperlipidaemia, n (%)	25 (36.2)	15 (20.8)	0.066
CMR measurements			
LVEDV index (ml/m^2^)	84.0 (75.4, 92.3)	79.0 (69.0, 87.3)	**0.012**
LVESV index (ml/m^2^)	33.7 (28.3, 41.8)	27.0 (24.3, 31.5)	**<0.001**
LV ejection fraction (%)	59.5 (56.0, 64.0)	65.0 (63.0, 67.0)	**<0.001**
RVEDV index (ml/m^2^)	76.9 (68.4, 86.0)	77.6 (67.0, 85.2)	0.837
RVESV index (ml/m^2^)	28.5 (23.7, 34.5)	25.6 (21.9, 30.0)	0.053
RV ejection fraction (%)	62.6 ± 6.0	66.1 ± 4.3	**<0.001**

BSA: body surface area; EDV: end-diastolic volume; ESV: end-systolic volume; LV: left ventricular; RV: right ventricular. Values with a normal distribution are shown as a mean ± standard deviation while values without a normal distribution are shown as a median with an interquartile range. * 14% missingness; ^†^ 4% missingness.

**Table 2 diagnostics-11-01109-t002:** Predictors of abnormal CMR findings and differences in measurements between patients with normal and abnormal CMR scans.

Normal CMR Scan (n = 33, 45.8%)		Abnormal CMR Scan (n = 39, 54.2%)	*p*-Value
Clinical characteristics			
Age (years)	46.0 (35.0, 51.0)	54.0 (42.0, 60.5)	**0.045**
Males, n (%)	9 (27.3)	25 (64.1)	**0.004**
BMI	27.1(4.7)	27.3 (5.1)	0.851
BSA	1.85 (0.20)	1.95 (0.21)	**0.039**
Smoking, n (%) *	7 (24.1)	8 (24.2)	1
Hypertension, n (%) ^†^	3 (9.4)	9 (24.3)	0.188
Ectopy burden, n	11,668 (5283, 20,684)	14,363 (4731, 27,583)	0.739
NSVT, n (%)	5 (15.2)	11 (28.2)	0.297
Uncommon PVC origin	4 (12.9)	13 (38.2)	**0.041**
CMR measurements			
LVEDVi (ml/m^2^)	78.6 ± 9.0	89.8 ± 13.2	**<0.001**
LVESVi (ml/m^2^)	28.4 (25.3, 31.3)	39.6 (34.7, 45.6)	**<0.001**
LVEF (%)	63.4 ± 3.6	55.5 ± 5.8	**<0.001**
RVEDVi (ml/m^2^)	73.8 (67.3, 85.1)	79.2 (69.1, 86.8)	0.369
RVESVi (ml/m^2^)	27.3 (8.0)	30.8 (10.1)	0.111
RVEF (%)	64.5 ± 5.5	60.9 ± 6.0	**0.012**
LGE, n (%)	1 (3.0%)	21 (53.8%)	**<0.001**

BMI: body mass index; EDVi: end-diastolic volume index; ESVi: end-systolic volume index; LGE: late gadolinium enhancement; LV: left ventricular; NSVT: non-sustained ventricular tachycardia; RV: right ventricular. Values with normal distribution are shown as a mean ± standard deviation while values without a normal distribution are shown as a median with an interquartile range. * 14% missingness; ^†^ 4% missingness.

## Data Availability

The data presented in this study are available on request from the corresponding author. All the data are stored at the local research server and are not publicly available.

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
