# Peer review of "Cardiac Magnetic Resonance to Detect the Underlying Substrate in Patients with Frequent Idiopathic Ventricular Arrhythmias"

_diagnostics, 2021, doi:10.3390/diagnostics11061109_

Round 1

Reviewer 1 Report

I read with interest the manuscript by Nikolaidou et al. I think it adds new data supporting the use of cardiac MRI diagnosis of patients with idiopathic ventricular arrhythmia. I do have some questions/remarks though, which in my opinion, may improve the quality of the manuscript:

  1. Was ECG done beforehand or only echocardiography to determine lack of structural disease? Please clarify that in the text and abstract.
  2. What were the individual diagnoses grouped as NICM - please provide details.
  3. I would be cautious in diagnosis previous myocarditis solely on the base of scar in CMR - maybe it is early stage DCM of other origin or left-sided AC?
  4. Could you please clarify if there is any added diagnostic value of FT in relation to LGE? It was not included in the multivariable model.
  5. Table 1 -  you state that LVEF was within normal range in patients - line 159, but in the table the lowest quartile starts with 56% (it is already borderline in CMR)
  6. Limitations - please state that mapping techniques (parametric imaging) were not used.
  7. Reference 9 seems started on the reference list.

Author Response

Thank you for your constructive comments which have been very helpful in improving our manuscript. We have incorporated nearly all of your suggestions and we believe that your comments have significantly improved the manuscript. We have uploaded a revised manuscript with tracked changes.
Point-by-point response to comments:
1) Was ECG done beforehand or only echocardiography to determine lack of structural disease? Please clarify that in the text and abstract.
Thank you for your comment. Indeed, an ECG was available for all patients and was used to exclude possible structural heart disease, together with the echo findings. This has been added in the text (lines 82, 83) and in the abstract (line 26).
2) What were the individual diagnoses grouped as NICM - please provide details.
Details about the possible cardiac diseases grouped as NICM have been added (lines 188-190).
3) I would be cautious in diagnosis previous myocarditis solely on the base of scar in CMR - maybe it is early stage DCM of other origin or left-sided AC?
We completely agree with this comment and we considered that in our diagnosis. To make it more clear for the reader, we changed the diagnosis to “possible previous myocarditis” (lines 133, 183, 284, 302), and added a comment in the limitations (lines 389-392).
4) Could you please clarify if there is any added diagnostic value of FT in relation to LGE? It was not included in the multivariable model.
Thank you for this comment. We did not include LGE in the multivariate model, because LGE was present only in patients and not in healthy volunteers, so we entered in the multivariate model the parameters that were present in both groups, patients and healthy volunteers. Also, there was no statistically significant difference in LV-GRS and LV-GCS (which were found to be significantly different in the different patient groups) in patients with and without LGE (please see following figure). A comment on the correlation of strain with LGE was added in lines 300-301 to clarify this.
5) Table 1 - you state that LVEF was within normal range in patients - line 159, but
in the table the lowest quartile starts with 56% (it is already borderline in CMR)
In the study, we included patients reported to have normal ejection fraction on the
echocardiograms performed by their referring physicians (as stated in lines 84-86). In
some patients, the LVEF was indeed borderline or mildly reduced on CMR. This
finding underlines the value of CMR on this patient group with frequent ventricular
ectopics, when performed after pharmacological suppression of ectopics with
procainamide. This finding also shows the limitations of transthoracic
echocardiography in the clinical practice (line 328-329).
6) Limitations - please state that mapping techniques (parametric imaging) were not
used.
A comment has been added in the limitations section (lines 392-394).
7) Reference 9 seems started on the reference list.
We apologise for this. It has been changed.

Reviewer 2 Report

The Authors presented an article "Cardiac Magnetic Resonance to detect the underlying substrate in patients with frequent idiopathic ventricular arrhythmias"

Specific comments:

1) You have used procainamide prior to CMR scanning. However, no references on the previous works, its function for this analysis are not presented. I think you should add some more discussion and safety data from the Table S3 footnote. So I think much more discussion and information relative to a dose of the drug and Its safety/toxicity should be presented. Some works about its safety in a dose-dependent manner should be presented and cited. 

Further side effects have been reported for procainamide; have any of those been observed? Please comment.

2) A quantitative assessment of the improvements of CMR image quality is missing. How the observed improvement in image quality influenced the diagnosis in the patients enrolled?

Author Response

Thank you for your constructive comments which have been very helpful in improving our manuscript. We have incorporated nearly all of your suggestions and we believe that your comments have significantly improved the manuscript. We have uploaded a revised manuscript with tracked changes.
Point-by-point response to comments:
1) You have used procainamide prior to CMR scanning. However, no references on the previous works, its function for this analysis are not presented. I think you should add some more discussion and safety data from the Table S3 footnote. So I think much more discussion and information relative to a dose of the drug and Its safety/toxicity should be presented. Some works about its safety in a dose-dependent manner should be presented and cited.
In the Materials and Methods section (CMR protocol), we have added information about the mechanism of action, the dose of procainamide and therapeutic plasma concentrations (lines 112-124). Further information about its mechanism of action and side effects/toxicity have been described in our recent article “Bolus Intravenous Procainamide in Patients with Frequent Ventricular Ectopics during Cardiac Magnetic Resonance Scanning: A Way to Ensure High Quality Imaging. Diagnostics (Basel) 2021,11(2); doi: 10.3390/diagnostics11020178”, as cited in the article.
Also, more information about the dose of procainamide administered presented in Table S3, was added in the Results section (lines 214, 215).
Further side effects have been reported for procainamide; have any of those been observed? Please comment.
The effects of procainamide on haemodynamic parameters (also described in Table S3) were added in the results section. Comment about the observed side-effects and other possible, but not observed side-effects were also added (lines 215-219).
2) A quantitative assessment of the improvements of CMR image quality is missing. How the observed improvement in image quality influenced the diagnosis in the patients enrolled?
As presented in Table S3, a 1 to 4 scale system was used to grade the image quality. Image quality and diagnostic capability significantly improved, as added in the text (lines 223-225).
